# Ras Suppressor-1 (RSU1) in Cancer Cell Metastasis: A Tale of a Tumor Suppressor

**DOI:** 10.3390/ijms21114076

**Published:** 2020-06-07

**Authors:** Maria Louca, Triantafyllos Stylianopoulos, Vasiliki Gkretsi

**Affiliations:** 1Cancer Biophysics Laboratory, Department of Mechanical and Manufacturing Engineering, University of Cyprus, 1678 Nicosia, Cyprus; louca.maria@ucy.ac.cy (M.L.); tstylian@ucy.ac.cy (T.S.); 2Biomedical Sciences Program, Department of Life Sciences, School of Sciences, European University Cyprus, 1516 Nicosia, Cyprus

**Keywords:** cell-extracellular matrix adhesion, actin cytoskeleton, invasion, migration, metastasis, breast cancer, hepatocellular carcinoma, glioblastoma

## Abstract

Cancer is a multifactorial disease responsible for millions of deaths worldwide. It has a strong genetic background, as mutations in oncogenes or tumor suppressor genes contribute to the initiation of cancer development. Integrin signaling as well as the signaling pathway of *Ras* oncogene, have been long implicated both in carcinogenesis and disease progression. Moreover, they have been involved in the promotion of metastasis, which accounts for the majority of cancer-related deaths. *Ras Suppressor-1 (RSU1)* was identified as a suppressor of *Ras-*induced transformation and was shown to localize to cell-extracellular matrix adhesions. Recent findings indicate that its expression is elevated in various cancer types, while its role in regulating metastasis-related cellular processes remains largely unknown. Interestingly, there is no in vivo work in the field to date, and thus, all relevant knowledge stems from in vitro studies. In this review, we summarize recent studies using breast, liver and brain cancer cell lines and highlight the role of RSU1 in regulating cancer cell invasion.

## 1. Introduction

Cancer is a multifactorial disease with a strong genetic component, as mutations in oncogenes or tumor suppressor genes significantly contribute to the initiation and development of tumors [1,2]. Although tumor formation is crucial and should be closely monitored and treated, most cancer patients do not die of the primary tumor but rather of the subsequent metastasis of cancer cells. Metastasis is a complex multistage process during which cancer cells dissociate from cell-extracellular matrix (ECM) adhesions, lose contact with their neighboring cells and finally detach from the primary tumor. Then, they degrade the surrounding ECM to invade adjacent tissues and are transported through the circulation or the lymphatic system to other distant organs, where they extravasate, adhere to the new environment and establish a new colony of malignant cells [3,4,5,6]. Notably, certain cancer cell types seem to show a preference with regard to their metastatic sites, a phenomenon known as metastatic tropism [7,8]. Breast cancer cells, for instance, tend to form metastases to the bones, the lungs, the liver and the brain, while prostate cancer cells tend to metastasize more towards the bone and pancreatic cancer cells show a preference to the liver and the lungs [9].

For the metastatic process to take place, integrins and integrin-related protein complexes formed at cell-ECM adhesion sites (also known as focal adhesion sites) are of fundamental importance [10,11,12]. Thus, upon integrin activation, a signaling cascade is initiated resulting in important changes in terms of cell behavior that affect cell survival and apoptosis, cell differentiation, proliferation and adhesion [13,14,15,16]. Moreover, due to the fact that most focal adhesion proteins maintain a tight connection either directly or indirectly with actin cytoskeleton, integrin activation also affects processes such as cell migration and invasion of surrounding matrix, which are both intrinsically linked to metastasis [17,18].

The Formation of ILK-PINCH-PARVA (IPP) Complex at Cell ECM Adhesion Sites

Integrin-linked kinase (ILK) is an important component of focal adhesions [19]. It was initially described as an intracellular serine/threonine protein kinase that interacts with the integrin β1 cytoplasmic domain [20] to modulate various cellular functions. However, increasing data indicate that in most cases, ILK acts as a pseudokinase given that it contains a domain with kinase homology that serves as a mediator of several protein–protein interactions, rendering ILK a scaffold protein at focal adhesions. Through its property to form protein-protein interactions, ILK has been shown to form a stable ternary protein complex at focal adhesions, being bound to alpha-parvin (PARVA) and PINCH-1 (Particularly Interesting new cysteine-histidine rich protein), which is also known as LIM Zinc Finger Domain Containing 1 (LIMS1) [21], thus forming the so-called ILK-PINCH-PARVA complex or IPP complex. In fact, it has been demonstrated that ILK is critically involved in the IPP protein complex formation and it is also responsible for targeting the IPP complex to focal adhesions [22,23]. The IPP complex, in turn, has been implicated in the regulation of several cell-ECM adhesion-mediated signaling pathways and many fundamental cellular functions, such as cell survival, cell differentiation and cell adhesion to the ECM, ensuring normal tissue homeostasis [21,24,25,26,27,28]. Moreover, ILK was found to play a vital role in promoting the aggressiveness of cancer cells by regulating the level and activation of several key molecular pathways downstream of integrins, such as PKB/Akt, Extracellular Regulated Kinase (ERK) and Glycogen synthase kinase-3β (GSK3β) [29,30,31]. Notably, the other members of the IPP complex also play crucial roles in regulating cell shape, spreading [32,33,34] and cell motility [35,36], cell survival [37], cell proliferation, apoptosis [38] and differentiation. For instance, PARVA is an important component of the IPP complex and has been found to regulate cell attachment and spreading through activation of Ras-related C3 botulinum toxin substrate 1 (Rac1) [37], while promoting cell survival through activation of Akt/PKB pathway [39]. Furthermore, PINCH-1, the other component of the IPP complex, has been considered to be a pro-survival gene that is also involved in regulating cell shape, morphology and motility [38,40].

Given the fact that hundreds of proteins assemble at the focal adhesion sites and make the cell’s adhesome [41], most of the focal adhesion proteins also serve as adaptor proteins for the attachment of more related proteins that altogether transmit signals from the external environment of the cell to the inner compartment. Thus, identification of accessory proteins that are able to associate with the IPP complex and modulate tissue-specific processes is essential to enhance our understanding of the focal adhesions and their involvement in health and disease.

In that regard, PINCH-1 has been also found to interact with another focal adhesion protein known as Ras Suppressor 1 (RSU1), and regulate cell survival, migration and spreading [28,32]. In fact, this interaction was further corroborated by data from a novel two-dimensional (2D)-gel electrophoresis analysis, known as interactions by 2D Gel Electrophoresis Overlap (iGEO) [42]. iGEO used affinity tags to PINCH-1 sites and expressed them both in vitro and in vivo in Drosophila. Affinity purification and mass spectrometry analysis followed which confirmed a core complex consisting of PINCH-1, RSU1, ILK and PARVA.

## 2. RSU1 in Normal Tissues

Among all other focal adhesion proteins that connect integrin signaling with intracellular signaling and actin cytoskeleton, RSU1 is of particular interest, as it was originally identified as a suppressor of *Ras*-dependent oncogenic transformation [43,44]. Ras family of proteins comprise a fundamental part of cellular signaling and are responsible for regulating cell survival, growth and differentiation. Moreover, missense mutations in several of the *Ras* genes have been linked to oncogenic transformation and are present in several types of cancer [45]. Therefore, a suppressor of *ras* oncogenes has, by definition, a great potential for anti-cancer therapy.

RSU1 is a 33 kDa protein, consisting of 277 amino acids (NCBI Reference sequence: NM_012425.3) [43,44] forming a series of leucine-based repeats having high extent of homology with leucine repeats found in the region of adenylyl cyclase that is responsible for *ras* activation in yeast [46]. RSU1 is encoded by *RSU1* gene located on the short arm of human chromosome 10 (10p13) [46]. Apart from the original RSU1 protein, another 29 kDa isoform (namely, RSU1-X1, with NCBI Reference sequence: XM_005252552.4) produced by alternative splicing has been reported to be present in more aggressive gliomas [47] and breast cancer cells [48].

Notably, RSU1 was also shown to localize to focal adhesions through its interaction with the LIM5 domain of PINCH-1 [49,50]. In fact, using unbiased screens and multiparametric image analysis of focal adhesions following siRNA-mediated silencing, RSU1 was identified as an important protein in cell’s adhesome [51,52]. Studies where *RSU1* was transiently overexpressed in NIH3T3 fibroblasts and PC12 pheochromocytoma cells revealed that RSU1 affected several kinases downstream of the *Ras* oncogene that are necessary for oncogenic transformation [53]. Specifically, *RSU1* overexpression inhibited c-Jun N-terminal Kinase (JNK), and activated ERK in response to Epidermal Growth Factor (EGF) [53]. Moreover, PC12 cells overexpressing *RSU1* exhibited significant growth inhibition through elevation of Cyclin Dependent Kinase Inhibitor 1A (p21) expression without being compromised in terms of their differentiation potential as seen by the fact that *RSU1* overexpression resulted in Nerve Growth Factor (NGF)-induced differentiation through ERK activation [54].

Interestingly, the connection of RSU1 with PINCH-1 seems to be crucial for its function, as depletion of *PINCH-1* reduces *RSU1* expression leading to increased JNK activity in primitive endoderm cells [55]. Similarly, RSU1 and PINCH-1 have been shown to regulate JNK signaling and contribute to epithelial sheet migration during dorsal closure in Drosophila melanogaster development [56]. In fact, it was later shown in Drosophila that RSU1 compensates for the loss of function occurring when the binding of PINCH-1 to ILK is compromised maintaining the organism’s viability and stabilizing the IPP complex [57]. To add more to the connection of RSU1 to PINCH-1, it was shown that in MCF10A mammary epithelial cells, RSU1 regulates PINCH-1 levels and stabilizes it, while the two proteins seem to act synergistically to regulate cell spreading through activation of Rac1 [32].

However, it should be noted that RSU1 function is not entirely dependent upon PINCH-1 localization to focal adhesion sites, as the depletion of either *PINCH-1* or *RSU1* resulted in decreased cell adhesion, migration and loss of actin stress fibers in MCF10A cells, while the reconstitution of *RSU1-*depleted cells with PINCH1-binding defective Rsu1 mutant rescues spreading and p38 activation [58,59]. This PINCH-1-independent function of RSU1 is presumed to be mediated by the truncated RSU1-X1 isoform, which has been previously shown in co-immunoprecipitation studies to be unable to bind to PINCH-1 [60]. Hence, RSU1 and PINCH-1 are necessary for regulating adhesion and migration through the IPP complex, but RSU1 is also connecting focal adhesions and spreading with Mitogen Activated Protein Kinase 14 (p38) signaling [58]. Furthermore, a recent study in non-transformed MCF10A human mammary epithelial cells revealed that RSU1 inhibits Akt phosphorylation and promotes the mRNA expression of tumor suppressor gene Phosphatase and Tensin homologue (PTEN) through p38 activation [61], ultimately leading to reduced survival, which reinforces its role as a growth regulator.

Finally, RSU1 has been implicated in basic cellular processes and functions of the central nervous system (CNS). More specifically, elimination of ILK in adult mammalian brain was found to enhance JNK activity and increased neural stem and progenitor cell proliferation via RSU1 loss, suggesting that RSU1 is critical in neurogenesis of the mammalian brain [62]. Furthermore, in another study, RSU1 regulated synapse maturation through preventing spontaneous clustering of extrasynaptic acetylcholine receptors in Caenorhabditis elegans [63], thus indicating a potentially crucial involvement in CNS physiology. RSU1 was also identified in an unbiased genetic screen for altered ethanol responses in Drosophila melanogaster as a potent regulator of ethanol consumption and data were confirmed in humans as well [64]. In fact, RSU1 was found to regulate reward-related phenotypes such as ethanol consumption both in flies and humans by connecting signaling from the integrins to the Rac1 small GTPase ultimately leading to modulation of synaptic plasticity [64].

A diagrammatic representation of RSU1 functions and known signaling pathways involved in regulating normal cell homeostasis is presented in Figure 1.

## 3. RSU1 in Tumor Tissues

Although the involvement of Ras proteins as GTPases in cancer progression through intracellular signaling transmission and actin remodeling has been well-established [65,66], and RSU1 was first characterized as a *Ras*-mediated oncogenic transformation suppressor, the exact role of *RSU1* in cancer is still vague [67]. Interestingly, while several studies have been performed in vitro using various cancer cell lines, an in vivo investigation of the role of RSU1 in cancer is still missing.

### 3.1. RSU1 in Breast Cancer

With regard to breast cancer, a study performed in 2000 by Vasaturo et al. [68] showed that overexpression of *RSU1* in MCF-7 breast cancer cells induced p21 activation and reduced cancer cell proliferation through inhibition of Cyclin-dependent kinase (CDK), proposing that RSU1 acts as a tumor suppressor. A more recent study, performed in breast cancer cell lines showed that *RSU1* is upregulated in more aggressive and highly invasive MDA-MB-231 breast cancer cells compared to the non-aggressive MCF-7 breast cancer cells, both at the mRNA and protein level, which indicates a deregulation in *RSU1* expression in the more cancerous cell line, perhaps as a compensatory mechanism to reduce cell proliferation rate. Interestingly though, when *RSU1* was silenced, *PINCH*-1 expression was upregulated and cell proliferation was enhanced through the inhibition of p53 and upregulation of a regulator of apoptosis, namely p53 Up-regulated Modulator of Apoptosis (PUMA) [69]. Interestingly, these results were further validated in 32 human breast cancer samples with or without metastasis to the lymph nodes having respective normal adjacent tissues as controls. *RSU1* was found to be dramatically and significantly elevated in metastatic breast cancer samples compared to non-metastatic and compared to the normal adjacent tissues and, in fact, its expression was shown to be negatively correlated with PINCH-1 expression and positively with PUMA expression [69].

Since all relevant in vitro studies were performed in two-dimensional (2D) culture systems, in which, by definition, cell-matrix interactions are not taken into account, a recent study developed three-dimensional (3D) culture models to better study the role of *RSU1* in a more physiologically relevant manner. In that regard, breast cancer cells were either grown inside a 3D collagen gel of tunable stiffness (by adjusting the collagen concentration) or were left to form tumor spheroids and were then embedded in 3D collagen gels in an attempt to investigate cancer cell invasion [48,70]. It was shown that *RSU1* was significantly upregulated in increased stiffness conditions, while its silencing diminished the invasive capacity of tumor spheroids through collagen gels. In fact, this was mediated by urokinase Plasminogen Activator (uPA) and Matrix metalloproteinase 13 (MMP13) [70].

Another recent study in breast cancer cells involved transient silencing of *RSU1* expression in two breast cancer cell lines and demonstrated that this silencing resulted in downregulation of Growth Differentiation Factor-15 (*GDF15*), a member of the Transforming Growth Factor-β (TGF-β) family of proteins, known to be associated with actin cytoskeleton reorganization and metastasis [71,72,73]. *RSU1* silencing also inhibited the expression of actin-modulating genes, namely *PARVA*, *RhoA*, Rho associated kinase-1 (*ROCK-1*) and *Fascin-1.* Most importantly, this inhibitory effect was completely reversed by human recombinant GDF15 treatment, which also rescued the inhibitory effect of *RSU1* silencing on cell migration and invasion [74], further suggesting that GDF15 can compensate for *RSU1* loss.

Interestingly, regarding the alternatively-spliced *RSU1* isoform (*RSU1-X1*), it was shown to be expressed in human gliomas [47]. Depletion of this isoform from breast cancer cells has been also found to inhibit their migration, while inhibitor studies revealed that the MEK-ERK pathway regulates its expression [60]. This RSU-X1 isoform was also observed to be present in highly invasive MDA-MB-231 and MDA-MB-231-Lung Metastasis-2 (MDA-MB-231-LM2) breast cancer cells, but not in the less invasive MCF-7 cells [70]. In addition, a recent study [48], investigating the involvement of *RSU1* isoforms in cancer cell metastasis, utilized shRNA-mediated silencing to generate breast cancer cell lines that permanently lacked *RSU1. RSU1* depletion in the two cell lines had completely opposite effects on cell migration, cell invasion and tumor spheroid invasion in 3D collagen gels. While *RSU1* depletion from MCF-7 cells resulted in an impressive and complete abrogation of cell migration, cell invasion and tumor spheroid invasion, its depletion from MDA-MB-231-LM2 cells dramatically promoted all three pro-metastatic properties. At the same time, the shorter *RSU1-X1* isoform was upregulated, perhaps as a compensatory mechanism for the loss of *RSU1.* Remarkably, when the truncated *RSU1-X1* was also eliminated in the cells that were permanently lacking *RSU1*, RSU1-depletion-induced cell migration and invasion were significantly inhibited along with a concurrent reduction in *uPA* expression [48].

Furthermore, a connection between RSU1 and miR-409-5p has also been made, as RSU1 was confirmed to be directly targeted by this miRNA in breast cancer cell lines MCF-7 and MDA-MB-231. Specifically, in cells that had been previously treated with a lentivirus that inhibited miR-409-5p, siRNA-mediated silencing of *RSU1* promoted cancer cell proliferation and migration, indicating that the regulatory effect of miR-409-5p inhibition in breast cancer is achieved through the inverse upregulation of RSU1 [75].

In conclusion, studies in breast cancer cells clearly show that both RSU1 isoforms promote breast cancer cell migration and invasion in vitro but there is also a mechanism in place by which the truncated *RSU1-X1* isoform acts as a back-up for performing the functions of *RSU1* when the latter is lost. Hence, ideally both isoforms should be blocked to effectively abolish the invasive and migratory potential of breast cancer cells (Figure 2).

### 3.2. RSU1 in Hepatocellular Carcinoma

Little is known regarding the role of RSU1 in hepatocellular carcinoma, with the thus far available data being in agreement with what has been shown in breast cancer. More specifically, *RSU1* expression was found to be dramatically elevated in more aggressive HepG2 hepatocellular carcinoma cells compared to the non-metastatic Alexander cells and its elimination promoted cell proliferation [76]. In addition, Hepatitis C virus infection was shown to upregulate *RSU1* expression promoting a cancerous phenotype [77]. Moreover, Donthamsetty et al. [78] also showed that elimination of PINCH-1 in mouse hepatocytes resulted in reduced *RSU1* expression, which in turn led to increased hepatocyte proliferation. The tumor suppressor role of RSU1 is further corroborated by the fact that *RSU1* is frequently deleted in hepatocellular carcinomas [79]. Finally, with regard to liver cancer cell invasion and similarly to breast cancer cells (Figure 2), depletion of the *RSU1* from aggressive hepatocellular carcinoma cells leads to significantly impaired cell invasion [76].

### 3.3. RSU1 in Glioblastoma

As *RSU1* has been previously linked to basic functions of the CNS [54,63], it is not surprising that it is also involved in the pathogenesis of glioblastoma, the most aggressive type of brain cancer [13,47,80]. Transient overexpression of *RSU1* in U251 glioblastoma cells that express low levels of *RSU1* reduced their growth rate in vivo and reduced aggressive cell behavior, again indicating that *RSU1* likely acts as a tumor-suppressor gene [46]. However, no information was available on the role of *RSU1* on basic metastasis-related properties, such as cell migration and invasion, until recently.

A recent study explored the role of *RSU1* in a panel of brain tumor cell lines and clearly showed that the more aggressive brain cells (A172 and U87-MG) exhibited dramatically increased expression of *RSU1* both at the mRNA and protein level in contrast to the less aggressive brain cell lines (H4 and SW1088), which express the gene at minimal levels. Interestingly, *RSU1* was shown to behave differently in the various brain cell lines with regard to in vitro cell migration and invasion and did not show the uniform pattern seen in breast cancer cells, where *RSU1* promoted the in vitro metastatic properties of cells. On the contrary, *RSU-1* silencing was shown to inhibit migration and invasion of aggressive cells and promote those of less aggressive cells [80], indicating that *RSU1* promotes the invasion capacity of aggressive glioma cells A172 and U87-MG that express high levels of *RSU1.* This was achieved through activation of Signal Transducer and Activator of Transcription (STAT6) and MMP-13, while the inhibition of cell invasion in less aggressive H4 and SW1088 glioma cells, which express *RSU1* in low levels, was also observed to take place through negative regulation of STAT6 and MMP13 [80].

Thus, *RSU1* apparently has distinct roles with regard to glioblastoma cell invasion depending on the cells’ aggressiveness as well as based on its expression level in the specific cells (Figure 2). In more aggressive glioma cells in which *RSU1* is elevated, cell invasion is promoted, while in less aggressive cells with low *RSU1* expression, it is inhibited [80]. The molecular mechanism by which this is achieved is not yet defined, but it is in accordance with other focal adhesion proteins whose level is also associated with cell migration capacity [81]. It is also reminiscent of the TGF-β [82] and GDF15 [83] mechanisms of action, which are known to act as tumor suppressors during early stages of the disease and as oncogenes at later stages. In fact, it was recently shown that the link between RSU1 and GDF15 is active in brain cells similarly to what happens in breast cancer cells, in regulating cell aggressiveness [84], as GDF15 is known to be associated with cancer cell malignancy and is elevated in glioblastoma patients [85]. Furthermore, the correlation of the expression levels of GDF15 and RSU1 determines the aggressiveness of brain cells through the regulation of RhoA, PINCH-1 and MMP13 [84], providing the basis for future investigations towards deciphering the molecular mechanism of RSU1 action.

A summary of existing studies on the role of RSU1 in cancer development and progression is presented in Table 1 below.

## 4. Current Clinical Knowledge

Although there is a lack of in vivo work related to RSU1 function, there is significant evidence from clinical samples corroborating the in vitro findings. Specifically, analysis of Kaplan-Meier survival plots from human breast cancer patients revealed that high *RSU1* expression is associated with poor prognosis for distant metastasis-free survival and remission-free survival [68,70]. Furthermore, protein expression analysis data from 23 human breast cancer samples showed that RSU1 is elevated in metastatic breast cancer cells, while the levels of the truncated isoform, *RSU1-X1,* are significantly reduced [48]. This is also in accordance with in vitro data in breast cancer cells lines, where more metastatic cell lines express RSU1 at higher levels [48,69], and further supports the hypothesis that RSU1 promotes a metastatic phenotype.

Regarding brain cancer, the first report on the involvement of RSU1 in glioblastoma was made as early as in 1995, showing that the *RSU1* gene is frequently deleted in high-grade gliomas [46], but no other evidence is available in human samples thus far.

## 5. Conclusions

In summary, RSU1 is a focal adhesion protein that seems to play a tumor suppressor role in breast cancer [48,69,74], liver cancer [76] and glioblastoma [80,84], although its involvement in the regulation of cell migration and invasion in vitro in breast, liver and brain cancer cells also suggests cell-type specific metastatic promoting functions. In aggressive liver and breast cancer cells, *RSU1* is upregulated, and the blocking of its expression efficiently inhibits cell migration and invasion. This suggests that RSU1 could be a therapeutic anti-metastatic target in liver and breast cancer. In the case of breast cancer cells, which express both RSU1 isoforms, the expression pattern of this isoforms should be determined for appropriate therapeutic targeting. In brain cancer cells, there is only one RSU1 isoform expressed, but its mode of action depends on its expression level. In aggressive brain cancer cells, which express *RSU1* at high levels, RSU1 promotes cell invasion and migration, and thus, inhibiting it would be therapeutically beneficial. However, in non-aggressive brain cancer cells that express RSU1 at low levels, inhibition of RSU1 is not advised, as it has the opposite outcome.

Several aspects on the role of RSU1 in human cancer remain unknown and need to be addressed in the future. Firstly, *RSU1* expression and function should be evaluated in additional tumor types and human tumor samples. Secondly, in vivo studies are of utmost importance to further strengthen the conclusions made from the in vitro experiments. Thirdly, the presence of the truncated RSU1 isoform should be further investigated in other cancer types and its interplay with the full length RSU1 needs to be better defined. Lastly, GDF15 seems to be an important mediator of RSU1 functions both in breast cancer and in glioblastoma, but the molecular mechanism of its action, its interaction with RSU1 and possible modulators of that interaction are still unknown. 

We have presented here some evidence on the significance of RSU1 in cancer cell invasion and metastasis in breast cancer, liver cancer and glioblastoma. Future studies will shed some light on the potential of RSU1 as a therapeutic target against metastatic cancer.

## Figures and Tables

**Figure 1 ijms-21-04076-f001:**
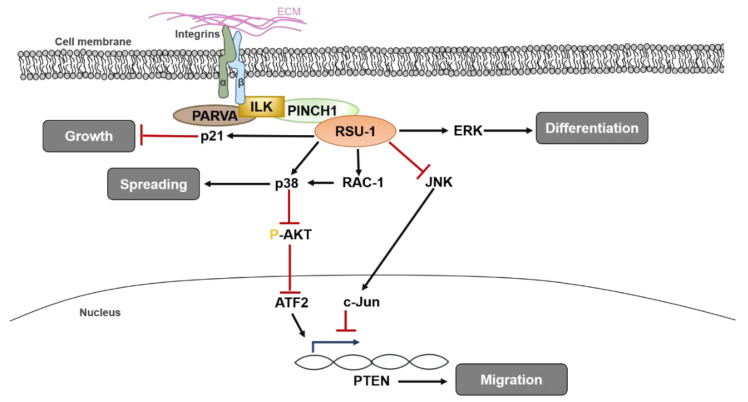
Diagrammatic representation of the role of RSU1 in normal tissues, where it regulates fundamental cellular processes such as spreading, migration, differentiation and proliferation of normal cells. ATF2: Activating Transcription Factor-2, ERK: Extracellular Regulated Kinase, ILK: integrin Linked Kinase, JNK: c-Jun N terminal kinase, PARVA: alpha parvin, PINCH-1: Particularly Interesting New Cysteine-Histidine rich protein, PTEN: Phosphatase and Tensin homologue, RAC-1: Ras-related C3 botulinum toxin substrate 1.

**Figure 2 ijms-21-04076-f002:**
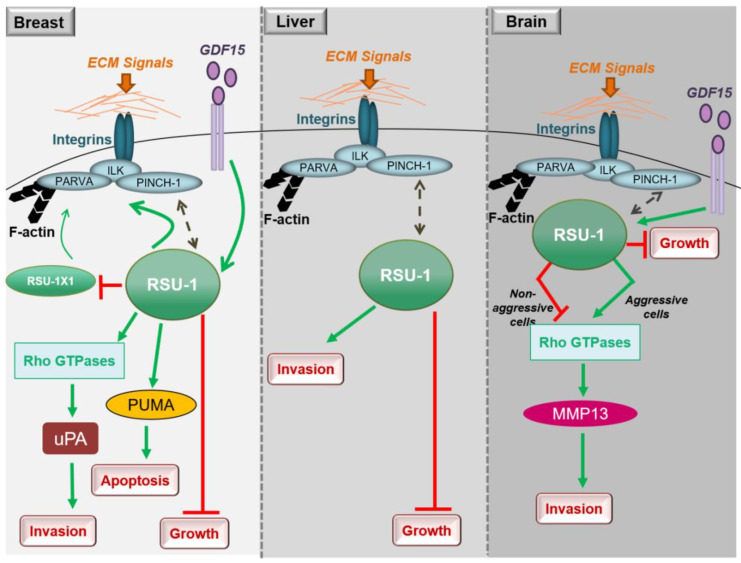
Diagrammatic representation of the role of RSU1 with regard to the cancer-related properties of breast, liver and brain cancer cells.

**Table 1 ijms-21-04076-t001:** Summary of studies on the role of RSU1 in cancer.

Cancer Type	Cell Lines	RSU1 Role	References
Breast	MCF-7 MCF-7 and MDA-MB 231 MCF-7 and MDA-MB-231 MDA-MB-468	Reduces proliferation Induces apoptosis Induces invasion Reduces migration	[68] [69] [48,70,74] [60]
Liver	HepG2	Reduces proliferation	[76]
Glioblastoma	U251 H4 and SW1088 A172 and U87-MG	Reduces proliferation Reduces invasion and migration Induces invasion and migration	[46,47] [80,84] [80,84]

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
