# Peer review of "Ras Suppressor-1 (RSU1) in Cancer Cell Metastasis: A Tale of a Tumor Suppressor"

_ijms, 2020, doi:10.3390/ijms21114076_

Round 1

Reviewer 1 Report

 The manusript has been accepted.

Author Response

We are grateful to the reviewer for critically evaluating our revised manuscript.

Reviewer 2 Report

Louca et al review information about the focal adhesion protein RSU1 in normal processes and cancer. Although there are still a lot of gaps in our understanding of the functions of RSU1, by summarizing the current knowledge the readers will at least get a good overview of what is already known and what is not.

The review is nicely written and includes novel literature. Yet, given the scarcity of information about RSU1, many questions remain that cannot be answered by reviewing the current literature. Thus, at many points the authors could guide readers by adding some more context instead of simply highlighting the findings of recent articles.

Despite the very nice work done by the authors in this review, by reading the text the readers may not really appreciate why it is necessary to review the knowledge about this protein. This point needs to be improved.

In addition, addressing these few minor points would further strengthen the manuscript:

  1. At the beginning of chapter 2, it would be helpful to stress out why RSU1 is such an important player that it merits an entire review. In addition, some extra information about the RSU1 protein (domains, motifs, etc.) would be helpful.
  2. How does RSU1 perform PINCH1-independent roles?
  3. The cellular processes in which RSU1 is implicated should be explained in more detail. For instance, it seems that its role in the CNS is responsible for the regulation of reward-related phenotypes (Ojalde et al PNAS 2015).
  4. The opposite role as oncogene or tumor suppressor need to be explained better. How do low levels prevent tumorigenesis whereas high levels act oncogenic? What are the mechanistic differences?
  5. The functional consequences of the information given in this manuscript for future cancer treatment need to be expanded. Would it be a good choice to inhibit its expression? Can it be a tumor target? If so, how could it be targeted in human cancer? Which adverse effects could be expected? Are all its tumor-promoting effects mediated via RAC1-signaling? Could patients with high expression levels be especially susceptible to RAC1-targeted therapies?
  6. Is it really a direct activity mediated by RSU1 that is responsible for its pro-tumorigenic role, or is it rather its effect on focal adhesions?

Author Response

Point by point response to the reviewer’s comments

We thank the reviewers for their constructive criticism and helpful comments and recommendations, which we have fully addressed in the revised manuscript. Below, please find a point-by-point response to the reviewers’ comments in blue colour and italics while new edits in the revised manuscript are also shown in blue.

Reviewer’s comments

Louca et al review information about the focal adhesion protein RSU1 in normal processes and cancer. Although there are still a lot of gaps in our understanding of the functions of RSU1, by summarizing the current knowledge the readers will at least get a good overview of what is already known and what is not.

The review is nicely written and includes novel literature. Yet, given the scarcity of information about RSU1, many questions remain that cannot be answered by reviewing the current literature. Thus, at many points the authors could guide readers by adding some more context instead of simply highlighting the findings of recent articles.

We thank the reviewer for critically reading our manuscript and providing constructive comments.

Despite the very nice work done by the authors in this review, by reading the text the readers may not really appreciate why it is necessary to review the knowledge about this protein. This point needs to be improved.

We agree with the reviewer and proper modifications have been made in the revised manuscript (please see 1st paragraph of section 2, lines 98-104).

In addition, addressing these few minor points would further strengthen the manuscript:

  1. At the beginning of chapter 2, it would be helpful to stress out why RSU1 is such an important player that it merits an entire review. In addition, some extra information about the RSU1 protein (domains, motifs, etc.) would be helpful.

Very nice and valid comment. Following the reviewer’s suggestion, we added a paragraph at the beginning of chapter 2 of the revised manuscript explaining why studying RSU1 and writing a review about it is important (lines 98-104). Additional information on the motifs involved in the protein has been also included in the revised version of the manuscript (please see 2nd paragraph of section 2, lines 105-111).

    2. How does RSU1 perform PINCH1-independent roles?

Very important comment. We have added this piece of information in the revised version of our article (please see lines 138-140).

    3. The cellular processes in which RSU1 is implicated should be explained in more detail. For instance, it seems that its role in the CNS is responsible for the regulation of reward-related phenotypes (Ojalde et al PNAS 2015).

We are grateful to the reviewer for this comment as well, and we have edited the revised manuscript accordingly (please see lines 153-157).

  4.The opposite role as oncogene or tumor suppressor need to be explained better. How do low levels prevent tumorigenesis whereas high levels act oncogenic? What are the mechanistic differences?

Following the reviewer’s suggestion, an explanation is provided for the oncogene/tumor suppressor role of RSU1 (please see lines 279-283 of the revised manuscript.

    5. The functional consequences of the information given in this manuscript for future cancer treatment need to be expanded. Would it be a good choice to inhibit its expression? Can it be a tumor target? If so, how could it be targeted in human cancer? Which adverse effects could be expected? Are all its tumor-promoting effects mediated via RAC1-signaling? Could patients with high expression levels be especially susceptible to RAC1-targeted therapies?

As explained in lines 323-334, there is no uniform answer to cover all types of cancers. RSU1 can be a therapeutic target but a more thorough analysis in more cell lines and tumor samples needs first to be performed. Regarding Rac signaling, please see lines 349-351 where this dimension has been added in the conclusions’ section of the revised manuscript.

 6. Is it really a direct activity mediated by RSU1 that is responsible for its pro-tumorigenic role, or is it rather its effect on focal adhesions?

Following the reviewer’s interesting suggestion, we have included this notion in the conclusion’s section of the revised manuscript (please see lines 339-340).
